# Improving Radiology Report Generation Quality and Diversity through Reinforcement Learning and Text Augmentation

**DOI:** 10.3390/bioengineering11040351

**Published:** 2024-04-03

**Authors:** Daniel Parres, Alberto Albiol, Roberto Paredes

**Affiliations:** 1Campus de Vera, Universitat Politècnica València, Camí de Vera s/n, 46022 Valencia, Spain; alalbiol@iteam.upv.es (A.A.); rparedes@prhlt.upv.es (R.P.); 2Valencian Graduate School and Research Network of Artificial Intelligence, Camí de Vera s/n, 46022 Valencia, Spain

**Keywords:** radiology report generation, reinforcement learning, text augmentation, machine learning, deep learning, vision transformer, chest X-rays, medical image, text generation

## Abstract

Deep learning is revolutionizing radiology report generation (RRG) with the adoption of vision encoder–decoder (VED) frameworks, which transform radiographs into detailed medical reports. Traditional methods, however, often generate reports of limited diversity and struggle with generalization. Our research introduces reinforcement learning and text augmentation to tackle these issues, significantly improving report quality and variability. By employing RadGraph as a reward metric and innovating in text augmentation, we surpass existing benchmarks like BLEU4, ROUGE-L, F_1_CheXbert, and RadGraph, setting new standards for report accuracy and diversity on MIMIC-CXR and Open-i datasets. Our VED model achieves F_1_-scores of 66.2 for CheXbert and 37.8 for RadGraph on the MIMIC-CXR dataset, and 54.7 and 45.6, respectively, on Open-i. These outcomes represent a significant breakthrough in the RRG field. The findings and implementation of the proposed approach, aimed at enhancing diagnostic precision and radiological interpretations in clinical settings, are publicly available on GitHub to encourage further advancements in the field.

## 1. Introduction

Radiology report generation (RRG) is a challenging task where the goal is to interpret radiographic images and generate detailed reports on potential patient pathologies. In contrast to typical computer vision (CV) tasks, which aim to identify objects in images, RRG focuses on diagnosing potential pathologies and determining their presence, absence, or uncertainty. Moreover, limited data availability and the diverse nature of medical reports pose significant challenges. This task is crucial for streamlining and enhancing patient care, as it enables the analysis of individuals’ health and the rapid detection of diseases. Furthermore, it serves as assistance and support for medical professionals. The RRG task parallels other works such as [1], where complex data are handled, and models trained from such data must ensure specific security criteria. This similarity arises from the need to develop models capable of diagnosing pathologies precisely, as in this type of problem, patients’ health is put at risk. Current approaches mainly rely on deep learning (DL), specifically the vision encoder–decoder (VED) architecture [2,3,4,5,6,7,8], incorporating components like memories [9,10] or reinforcement learning (RL) [5,11,12] to improve performance.

This study presents a two-stage VED pure-transformer architecture for chest RRG. In the first stage, conventional negative log-likelihood (NLL) training is employed, while the second stage focuses on RL optimization for various metrics. These metrics encompass embedding comparison for semantic coherence [13], entity and relationship graph generation for pathology descriptions [14], and NLL. Additionally, we propose the integration of text augmentation and hard negative mining techniques. This comprehensive approach surpasses current chest RRG methodologies, enhancing report variability, factual correctness, completeness, and overall model generalization.

The key contributions of this study are as follows:Harnessing vision encoder–decoder transformer frameworks and reinforcement learning to enhance radiology report quality, factual correctness, and completeness.Proposing text augmentation to the training workflow. This technique allows for leveraging the scarcity of data by generating new reports. This leads to an improvement in diversity, averaging a threefold increase compared to the state of the art.Due to the challenge of measuring the quality of generated reports, we focus on employing specialized natural language processing-based metrics to evaluate radiology reports comprehensively.Open-sourcing our methodology on GitHub at https://github.com/dparres/Diversifying-Radiology-Report-Generation (accessed on 1 April 2024) to propel progress in diagnostic precision and radiological interpretation within clinical settings.

### Related Work

In recent years, deep learning techniques have led various approaches to classify and detect pathologies in X-ray images. Notable efforts in this domain include Schlegl et al. [15], who proposed a convolutional network (CNN) for classifying tissue patterns in tomographies, utilizing semantic descriptions in reports as labels. Building on this success, subsequent neural network models were explored for X-rays, such as Shin et al. [16], who introduced a CNN for chest X-ray images and a recurrent network (RNN) for annotations, jointly trained to annotate diseases, anatomy, and severity. Other approaches, like that of Moradi et al. [17], focused on annotation through the concatenation of a CNN and an RNN block to identify regions of interest. Notably, Rubin et al. [18] employed parallelly trained CNNs for frontal and lateral chest X-ray views to estimate possible pathologies. The interest in RRG has grown, led by impactful models like TieNet [4], as well as innovative contributions from Li et al. [3], Jing et al. [19], and Jing et al. [19], each leveraging advanced techniques such as pretrained networks, coattention mechanisms, and retrieval policy modules for efficient disease classification and report generation.

Deep learning applications in radiology encounter obstacles due to limited and unstructured data availability, exacerbating the normal–abnormal case imbalance and complicated by ambiguous radiologist reports [20]. CNN models such as AlexNet [21], VGG-16 [22], GoogLeNet [23], and ResNet [24] dominate medical text–image mining, with the increasing prevalence of end-to-end training [20]. However, the model comparison is hindered by dataset diversity, although incorporating radiology reports is expected to grow [20]. Radiology reports, diverse in style and influenced by biases, pose challenges in training robust models, with few datasets meeting scalability and accessibility criteria [25]. Most RRG systems concentrate on X-ray tests, with emerging applications for CT scans and MRI datasets, each encountering unique challenges [25].

Alfarghaly et al. [26] introduced a novel technique for generating radiology reports from chest X-rays with DistilGPT2 [27]. This approach highlights faster training and improved metrics. Although limitations persist, larger datasets’ release is suggested for enhanced model generalization and critical evaluation of quantitative methods [26]. Yang et al. [28] presented a method combining general medical and specific medical knowledge with a multihead attention mechanism for chest RRG. Furthermore, Pan et al. [29] proposed a method for generating chest radiology reports through cross-modal multiscale feature fusion. This architecture aims to enhance location sensitivity and disease characterization. Additionally, these advancements pave the way for aligning scales and integrating knowledge graphs, enhancing the accuracy of report generation [29]. Yang et al. [30] introduced a highly accurate and automated radiology generation framework coupled with a novel automatic medical knowledge updating mechanism, enhanced by a multimodal alignment approach. Zhao et al. [31] presented a knowledge enhancement technique leveraging medical knowledge in dictionary form alongside historical knowledge, complemented by a multilevel alignment method to mitigate modal differences between text and image. Nicolson et al. [32] introduced CvT2DistilGPT2, leveraging a convolutional vision transformer for optimal encoder warm starting and highlighting GPT2’s [33] superiority in decoder warm starting over BERT [34]. Despite the proposals’ novelty, they all rely on metrics such as BLEU and ROUGE-L for comparison, which may not be entirely suitable for measuring the presence, absence, or uncertainty of pathologies. Furthermore, these metrics fail to consider the interrelation of pathologies within the report. Additionally, the monotony and repetition in generated reports are noteworthy factors attributed to the inherent difficulty of the RRG task.

Due to the increasing interest in automating RRG, specific architectures have been designed to address chest RRG tasks. Liu et al. [5] proposed a CNN for extracting visual features, followed by a sentence decoder and word decoder for generating report topics and composing the final report. The model is fine-tuned using RL with CheXpert [35] labels. Chen et al. [9] introduced the memory-driven transformer, employing relational and memory-driven layers to enhance information retention and incorporation into the decoder. Meanwhile, Chen et al. [10] presented a VED based on cross-model memory networks, leveraging shared memory to capture and utilize visual–textual alignments.

Liu et al. [6] and Liu et al. [7] proposed leveraging unsupervised construction of knowledge graphs to replicate radiologists’ patterns to generate reports. Focusing on RL, Miura et al. [11] suggested employing two metrics: one ensuring the generation of domain entities (${\mathrm{fact}}_{ENT}$) and the other maintaining coherent entity descriptions (${\mathrm{fact}}_{ENTNLI}$). These metrics are optimized alongside BERTScore [13], a semantic equivalence metric. Delbrouck et al. [12] propose a competitive chest RRG approach utilizing a VED. The model employs a DenseNet-121 [36] as the optical encoder and a single-layer BERT [34] as the decoder. Trained with RL, the optimization involves three metrics: RadGraph [14], BERTScore, and NLL. RadGraph, a neural network, constructs semantic annotations by forming a graph of entities and relationships in reports. The reward is computed by comparing the hypotheses’ entities and relationships against the reference, resulting in higher-quality reports surpassing values achieved by ${\mathrm{fact}}_{ENT}$ and ${\mathrm{fact}}_{ENTNLI}$. These proposals employ chest-RRG-oriented metrics to enable more precise comparisons of report quality compared to metrics such as BLEU or ROUGE-L. Despite this and using RL for training, the reports exhibit repetitiveness and a lack of diversity.

## 2. Materials and Methods

This section of the paper presents the primary approach for the chest RRG task. Firstly, the neural models are introduced, and we propose four different architectures to analyze the most suitable for the RRG task. Subsequently, the training workflow of the models is outlined. Then, the text augmentation technique is introduced to leverage the scarcity of data. The final subsection presents the databases to be utilized and the metrics for comparing models and measuring the quality of the generated reports.

### 2.1. Proposed Architectures

Our model architectures for addressing the chest RRG problem are VED models, with Swin [37] as the vision model and BERT [34] as the language model or decoder. We opted for Swin for its outstanding performance in various computer vision tasks such as classification, object detection, and semantic segmentation [37]. Compared to ViT [38], Swin’s hierarchical nature and more efficient attention mechanism based on shifted windows make it more suitable for X-ray analysis. Our models utilize both the base and small variants, denoted as SwinB and SwinS. These architectures are pretrained with ImageNet [39] and feature an input image size of three channels with a width and height of 224 pixels. They have architectural depths of 2, 2, 18, and 2 for each layer, with a patch size of 4. The primary difference lies in the number of heads, with SwinB utilizing sizes 4, 8, 16, and 32 and SwinS using 3, 6, 12, and 24.

For the decoder model, BERT was selected due to its proven effectiveness across various natural language processing (NLP) tasks [34,40,41]. However, we diverge from the standard configuration by employing only three layers with a hidden size of 1024, rather than the original 12 layers. Moreover, our investigation includes the analysis of two distinct decoders: one word-based, with a vocabulary of approximately 9.8k words, and another subword-based, featuring a vocabulary of 30k subwords. The integration of the vision model with the decoder model is facilitated through cross-attention layers, enabling the introduction of visual features extracted from the radiograph into the decoder. This integration allows for the autoregressive generation of reports.

Following this approach, the proposed VED models for chest RRG analysis are SwinB+BERT9k, SwinB+BERT30k, SwinS+BERT9k, and SwinS+BERT30k.

### 2.2. Training Workflows

In radiological contexts, multiple images (such as anterior–posterior and lateral views) are commonly generated per patient study. We employ a multi-image approach to address this, restricting it to three images per study. This involves concatenating features extracted by the Swin encoder for each image and processing them in the decoder. Furthermore, we apply random transformations—including translation, scaling, rotation, and adjustments to brightness and contrast—to augment training images.

Our training process comprises two stages, as illustrated in Figure 1. In the initial stage, the NLL loss function is employed, utilizing teacher forcing to train the model based on the generated hypothesis and the reference report. This stage typically encompasses approximately twelve training epochs. Subsequently, in the second stage, RL is introduced using the self-critical sequence training (SCST) algorithm [42]. SCST performs two forward passes of the VED: greedy decoding and beam-search multinomial sampling (sampling decoding). The first pass involves inference through greedy decoding to obtain the report $Y_{g}$ without calculating gradients. Meanwhile, the second pass employs sampling decoding, calculating gradients to obtain the sampling report $Y_{s}$.

RL aims to optimize specific metrics by using them as rewards. In our approach, we propose utilizing two metrics: BERTScore [13] and RadGraph [14]. BERTScore effectively improves grammar and semantics within models, while F_1_RG_*ER*_ aids the model in prioritizing pathology-related entities and relationships.

After obtaining the reports $Y_{g}$ and $Y_{s}$, we must calculate a loss for each metric to optimize ($Loss_{metric}$). This $Loss_{metric}$ is subsequently used to compute a weighted sum to obtain the final $Loss_{RL}$ and update the model weights. Equation (Equation 1) outlines the process to derive the $Loss_{metric}$, where the metric rewards are computed for $Y_{s}$ and $Y_{g}$ using the reference report $Y_{ref}$. These rewards are then subtracted and multiplied by the logarithm of the probabilities calculated during the generation of the $Y_{s}$ report.
(1)$$Loss_{metric}\left(Y_{s},Y_{g},Y_{ref}\right)=-\left(r_{metric}\left(Y_{s},Y_{ref}\right)-r_{metric}\left(Y_{g},Y_{ref}\right)\right)\log\left(\mathrm{Pr}\left(Y_{s}\right)\right)$$

Each $Loss_{metric}$ contributes as a distinct weighted term to the final RL loss, as presented in Equation (Equation 2). To obtain the $Loss_{RL}$ for updating the model weights, $Loss_{BERTScore}$, $Loss_{F_{1}RG_{ER}}$, and the same loss utilized in the initial training stage, the $Loss_{NLL}$, are summed. In our experiments, we set α=β=0.495 and γ=0.010 as the weighting factors.
(2)$$Loss_{RL}=\alpha Loss_{BERTScore}+\beta Loss_{F_{1}RG_{ER}}+\gamma Loss_{NLL}$$

### 2.3. Text Augmentation

Data augmentation (DA) is the most prevalent technique for addressing machine learning problems with limited data. This approach significantly contributes to enhanced generalization and improved data utilization. However, regarding RRG, the focus has primarily been on augmenting input images. To the best of our knowledge, no prior approach has investigated the impact of augmenting radiology reports, a technique known as text augmentation (TA). In chest RRG, reports are often drafted by different individuals in diverse manners, lacking specific templates, which results in significant variability inherent to natural language. Presently, state-of-the-art RRG algorithms encounter this data scarcity issue, often generating highly repetitive and monotonous reports.

In this study, we introduce a TA technique designed to enhance the quality and diversity of generated reports. Our TA approach involves splitting the reference report into phrases, considering the typical structure of reports with multiple sentences. These phrases are then randomly reorganized to create new reports while maintaining the original diagnosis, as depicted in Figure 1. This method mitigates overfitting concerns and promotes a more targeted learning of the mentioned pathologies. Furthermore, our TA significantly improves the quality, factual accuracy, and comprehensiveness of the generated reports, as demonstrated subsequently.

The TA technique described and utilized in this work is exclusively employed in the training workflow, specifically during the NLL and RL stages. This technique aims to leverage the reference reports to enhance the model training process and increase its robustness. The report sentences will be randomly ordered in each training epoch for each training sample.

### 2.4. Datasets and Metrics

To evaluate the competitiveness of our models, we employed two publicly available datasets: MIMIC-CXR [43] and Open-i [44]. Due to the relatively small size of the Open-i dataset, containing 3.3 k reports, it was exclusively utilized for testing. The MIMIC-CXR dataset comprises 152 k reports for training, 2.3 k for validation, and 2.3 k for testing. The workflow involves training with the 152 k reports, saving the model weights based on the best performance on the validation set, and finally using these weights to perform inference on both the MIMIC-CXR and Open-i test sets. It is important to note that in this study, the models were specifically designed to generate the findings section of the reports. Additionally, samples with empty findings sections were excluded from consideration.

Metrics play a crucial role in comparing models, with BLEU4 [45] and ROUGE-L [46] currently considered as the most widely used natural language generation (NLG)-oriented metrics for reports. Additionally, two chest-RRG-oriented metrics, specifically designed to evaluate the quality of radiology reports, have been employed: F_1_CheXbert (F_1_cXb) [47] and F_1_RG_*ER*_ [12,14]. These metrics leverage neural networks to assess the quality of generated reports, offering a higher semantic evaluation. F_1_cXb utilizes CheXbert, a transformer model capable of identifying the presence of the 14 CheXpert [48] pathologies (atelectasis, cardiomegaly, consolidation, edema, enlarged cardiomediastinum, fracture, lung lesion, lung opacity, pleural effusion, pneumonia, pneumothorax, pleural other, support devices, and no finding) in a hypothesis report, and calculates the F_1_-score based on the pathologies present in the reference report. On the other hand, F_1_RG_*ER*_ employs a transformer model to analyze reports, generating graphs for entities and relationships. It evaluates comparisons between entity and relationship structures in hypotheses and reference graphs to compute the F_1_-score.

## 3. Results

This section details the experiments carried out to develop models for chest RRG. We analyze the proposed architectures and then conduct an ablation study on effective training techniques and strategies. Finally, we assess the competitiveness of our proposal by comparing it with state-of-the-art models. All experiments were conducted using a computer equipped with an NVIDIA RTX 4090 GPU.

### 3.1. Analysis of Our Architectures

This study proposes four VED models, distinguished primarily by the number of trainable parameters in each component. Table 1 illustrates how SwinB+BERT30K is the largest model, while SwinS+BERT9k is the smallest. The remaining two models fall within a similar parameter range. Comparing architecture sizes is interesting because the RRG task operates in a data-limited environment due to the high variability of medical reports and their scarcity. Therefore, training models with high parameter counts can become challenging to tune efficiently. The number of parameters, along with the nature and size of the dataset, are crucial factors for effective learning that avoids underfitting and overfitting. Furthermore, the results presented in Table 1 encourage analysis of which components of the architectures may be more critical for obtaining meaningful reports.

In addition to the number of parameters, Table 1 showcases BLEU4, ROUGE-L, F_1_cXb, and F_1_RG_*ER*_ metrics. These metrics were obtained following the same workflow and hyperparameters across all four models. During the first stage, involving training with NLL, we conduct training for 12 epochs with a learning rate of $3\times 10^{-4}$. Subsequently, in the second stage, corresponding to RL training, we train for 15 epochs with a learning rate of $5\times 10^{-5}$. This learning rate value is crucial as it allows a good starting point for improving results based on previous training. The value of learning rates has been empirically explored using grid search with learning rate values of $5\times 10^{-3}$, $3\times 10^{-3}$, $1\times 10^{-3}$, $5\times 10^{-4}$, $3\times 10^{-4}$, $1\times 10^{-4}$, $5\times 10^{-5}$, $3\times 10^{-5}$, $1\times 10^{-5}$, $5\times 10^{-6}$, $3\times 10^{-6}$, and $1\times 10^{-6}$ for the first and second training stages. Values above $5\times 10^{-5}$ in RL training lead to highly repetitive reports, while values below it yield only marginal improvements, failing to achieve significantly better results than training with NLL. The training epochs are set to 12 and 15 for the first and second stages, respectively, proving sufficient to achieve satisfactory results. Increasing the epochs does not significantly improve report quality due to the task’s data scarcity and variability. During both stages, the learning rate is linearly decreased until reaching a value of zero at the end of the epochs. The results presented in Table 1 do not utilize image data augmentation, hard negative mining (HNM), or text augmentation.

Regarding report metrics on MIMIC-CXR, the SwinB+BERT9k model stands out, exhibiting superior performance compared to its closest counterpart, SwinB+BERT30k, and achieving a training time reduction of 16 h per epoch. For the SwinS models, results demonstrate a similarity, albeit with the BERT9k variant showcasing marginally better metrics across all parameters except F_1_cXb. Interestingly, F_1_cXb remains nearly identical between the two SwinS and SwinB models, indicating a direct impact of the encoder’s size on this metric’s competitiveness. Conversely, the choice between word-based or subword-based decoders does influence metrics such as BLEU4, ROUGE-L, and F_1_RG_*ER*_. Moreover, adopting word-based models yields superior metrics and significantly reduces training time, enhancing overall efficiency. The results from Open-i also rank the SwinB+BERT9k model as the most competitive and SwinS+BERT30k as the least competitive. Once again, this reaffirms that word-based models yield better results in the chest RRG task. Furthermore, the SwinS model demonstrates superior capability in generating reports by obtaining the best representations of radiographs.

Based on the results of the analysis, SwinB+BERT9k emerges as the most competitive and efficient model for chest RRG. However, various strategies can prove crucial in data-limited environments like chest RRG. This study specifically proposes using TA to leverage reports and significantly increase the training data. However, employing image augmentation techniques such as slight rotations, scale changes, shifts, random crops, and adjustments of brightness and contrast helps exploit the number of images. Moreover, since most datasets typically include more challenging samples than others, HNM is an option to consider. During training, samples with errors greater than the mean error plus their standard deviation are reutilized before moving to the next epoch.

Table 2 presents the ablation study of our best model, considering these strategies. The first row showcases the metrics achieved at the end of the NLL stage. The second row demonstrates how training with RL succeeds in improving these metrics; while image data augmentation shows a minor impact attributed to the robustness of transformers, HNM marginally enhances results by revisiting challenging samples at the end of each epoch. In contrast, TA emerges as a crucial technique, significantly improving chest-RRG-oriented metrics and BLEU4 by over one point. This underscores the significance and effectiveness of TA in efficiently leveraging the number of reports and furnishing the network with enhanced generalization capabilities.

### 3.2. Benchmarking with the State of the Art

To assess the competitiveness of our top model against other state-of-the-art models, we selected the proposals with the most competitive results on both the MIMIC-CXR and Open-i datasets, as detailed in Table 3. The table is divided into two sections: models trained with NLL and models trained with RL, comparing them across the same metrics presented in Table 2. Metrics optimized for models trained with RL are indicated in parentheses. The benchmark model we employ is SwinB+BERT9k utilizing RL and TA, as depicted in Table 2.

The results underscore the importance of utilizing RL for training, as models trained with NLL exhibit inferior performance. In NLL models for MIMIC-CXR, it can be observed that Nicolson et al. [32]’s proposal achieves the highest BLEU4 score while securing a second position in terms of ROUGE-L. Pan et al. [29]’s proposal obtains the highest ROUGE-L value. Regarding chest RRG metrics, due to their novelty, not all proposals have registered values, as image-captioning metrics such as BLEU4 or ROUGE-L are commonly used. Our proposal achieved the highest F_1_cXb values, followed by Delbrouck et al. [12]’s model; this trend was also observed for F_1_RG_*ER*_ values. When compared to the NLL models of the Open-i dataset, the highest BLEU4 score is achieved by Miura et al. [11], while Chen et al. [10] obtain the highest ROUGE-L score. Similar to MIMIC-CXR, in chest-RRG-oriented metrics, our model obtains the highest values, followed by Delbrouck et al. [12].

Table 3 also presents the results for methods trained with RL. The metric used within the RL reward is indicated in parentheses. In the MIMIC-CXR dataset, it is apparent that the BLEU4 score reported by Miura et al. [11] deteriorates in both models when compared to the one trained with NLL. However, it still manages to enhance the F_1_cXb score. Regarding Delbrouck et al. [12]’s models, there is an improvement of about one point in BLEU4 and ROUGE-L, with their model achieving the second-best scores for F_1_cXb and F_1_RG_*ER*_ at 62.2 and 34.7, respectively. Meanwhile, our SwinB+BERT9k model surpasses all RL models across all metrics, demonstrating superior quality, factual correctness, and completeness. In the case of the Open-i dataset, a similar trend is observed, with our model outperforming other proposals by approximately nine and six points in F_1_cXb and F_1_RG_*ER*_, respectively, compared to Delbrouck et al. [12]. The good results obtained in this database demonstrate the generalization and adaptability capacity of our training workflow to the chest RRG task.

Upon analyzing the results in Table 3, it becomes evident that the conventional approach to compare RRG systems relies on image-captioning metrics such as BLEU4 and ROUGE-L. As a result, all proposals rely on NLG-oriented metrics in their original works; however, the majority lack values for chest-RRG-oriented metrics due to their novelty. Nevertheless, NLG-oriented metrics may lack precision in clinical settings as they do not gauge syntax, semantics, or the comprehension of report meanings. Consequently, metrics tailored to the RRG task, like F_1_cXb and F_1_RG_*ER*_, have emerged to evaluate pathologies’ presence, absence, or uncertainty and their relationships within reports. These chest-RRG-oriented metrics offer a more accurate report quality assessment than NLG-oriented metrics. For instance, despite minimal differences in BLEU4 scores between Delbrouck et al. [12] and our approach in MIMIC-CXR, a significant gap is observed when measured with chest-RRG-oriented metrics. Another example of the imprecision of NLG-oriented metrics in assessing report quality can be seen in nearly identical BLEU4 scores between Miura et al. [11]’s models trained with NLL and RL in MIMIC-CXR, yet exhibiting a 12-point difference in F_1_cXb. High values in NLG-oriented metrics do not reflect the quality of a report, as neither BLEU4 nor ROUGE-L measures whether the mentioned pathologies are present, absent, or uncertain.

## 4. Discussion

This study presents an efficient workflow for training VED models explicitly designed for chest RRG. This workflow demonstrates remarkable performance in chest-RRG-oriented metrics. One pivotal finding lies in our analysis of the proposed architectures, where we show that utilizing the SwinB encoder yields superior radiograph representations. Notably, the word-based approach proves more competitive in the decoder component than its subword-based counterpart. This finding contrasts with strategies commonly employed in current large language models (LLMs). However, this discrepancy can be attributed to the inherent variability and scarcity of data in the chest RRG task. Furthermore, our ablation study underscores the critical role of TA in effectively leveraging data for this task, demonstrating substantial improvements in metrics over strategies like image augmentation or HNM.

A prevalent issue in RRG proposals is the high repetition and monotony frequently encountered in reports generated by other state-of-the-art approaches. Typically, models tend to converge to a local minimum, yielding nearly identical standardized reports across most patients. This phenomenon arises as deep learning algorithms settle at a midpoint where the generated reports broadly apply to most cases. These reports adhere to a template naming common pathologies in a fixed order and syntactic structure, irrespective of the patient. Consequently, this pattern results in reports of dubious quality, often overlooking less common pathologies, thereby undermining the reliability of diagnostic models.

Given the high repetition and monotony in the generated reports of state-of-the-art models, we propose an analysis of report diversity using N-grams in the MIMIC-CXR test set. This analysis allows for measuring diversity by examining the number of unique N-grams present in generated reports, as illustrated in Figure 2. Specifically, we compare reports generated by our model with those produced by the approach presented in Delbrouck et al. [12], acknowledged for being the most competitive proposal in the field, as outlined in Table 3. The graph illustrates a notably greater diversity in our generated reports compared to those of the Delbrouck et al. [12] model. Moreover, our model achieves a diversity difference averaging nearly three times their model.

Consequently, the TA strategy substantially augments the volume of reports for training and significantly boosts the performance of VED models, as demonstrated by chest-RRG-oriented metrics. An additional qualitative advantage of our TA-based approach lies in its capacity to generate less repetitive and more diverse reports compared to prior methods. We illustrate this with examples of reports from SwinB+BERT9k and their corresponding RadGraph entities in Table 4. ANAT-DP denotes anatomical body parts mentioned in the report, while OBS signifies observations associated with radiology images, categorized as definitely present (OBS-DP), definitely absent (OBS-DA), or uncertain (OBS-U). Despite improving report quality with techniques like TA, specific repetitions and monotony persist. This will need to be addressed using new metrics and strategies in the future.

In addition to the inherent complexity of this task, focusing on medical text generation necessitates the analysis and proposal of additional chest-RRG-oriented metrics based on NLP models. Exploring novel methods to assess the reference report against an estimated one can facilitate a more comprehensive evaluation of state-of-the-art models. Our study demonstrates the utility of F_1_cXb in verifying whether generated reports discuss pathologies present in the reference reports. Furthermore, F_1_RG_*ER*_ ensures that the present, absent, and unknown pathologies are correctly related in the reports. Considering these findings, these two metrics are essential for model comparison. Although these metrics effectively and efficiently enhance RL training, models frequently produce reports with incomplete sentences. This deficiency is evident in numerous state-of-the-art proposals, and despite the efficacy of TA, it still occurs occasionally. It is evidenced at the end of the third report in Table 4. This phenomenon arises from models learning that specific syntactic structures can benefit chest-RRG-oriented metrics like F_1_RG_*ER*_, even if they do not constitute syntactically correct sentences. Moving forward, we advocate for further research into novel metrics using alternative deep learning algorithms, such as LLMs like GPT-4 [50] or LLaMA [51].

Several aspects of the proposed approach for real-world RRG scenarios merit attention, including multilanguage support, standardization of reports, and improvements in picture archiving and communication systems (PACS). Since the decoder model primarily operates with English words, it encounters specific difficulties when used in other languages. Thus, it would be necessary to remove the last layer of the decoder model and adjust it based on the desired language; however, the rest of the model remains language-independent. Another challenge lies in the lack of standardization in report writing across different institutions, leading to variations in reporting guidelines. Consequently, RRG models can assist in integrating new reporting guidelines to standardize and unify databases for RRG. Furthermore, our approach can enhance PACS, which are commonly used in healthcare for securely storing and transmitting electronic images and clinical reports. Collaborating with expert radiologists, we can integrate RRG models to streamline processes within PACS, such as preparing automatic draft reports that radiologists can interactively correct. This approach can improve diagnostic efficiency and expert assistance times while enhancing the standardization and homogenization of reports.

The strengths and weaknesses of our model have been highlighted using different metrics. However, another essential aspect to analyze is the integration of the proposed workflow into the clinical setting. Given that the system relies on a VED deep learning model, its deployment only requires a computer with a GPU with at least 12 GB of memory. The model specializes in chest radiographs from different position views, such as anteroposterior, lateral, posteroanterior, and lateral decubitus. Therefore, it can integrate and analyze a wide variety of chest radiographs.

The scope of this work is focused on chest radiographs. However, our approach is not limited solely to its application in chest radiographs. The proposed approach is flexible and independent, allowing it to be applied to other radiological scenarios. Therefore, our approach can be seamlessly adapted for radiographs of different body parts. The only aspect requiring modification is the reward optimization during RL training. Since F_1_RG_*ER*_ primarily specializes in chest-related pathologies, this metric should be replaced with another relevant to the radiological scenario where our approach is intended for application. The remaining components, such as the encoder, decoder, TA, and decoding strategies, are independent of the radiological scenario. Despite the flexibility and independence of the approach, the major limitation for application to other radiological scenarios depends on the availability of training data. Acquiring large datasets to train VED models is crucial for improving the quality of reports and enhancing their applicability to different scenarios.

## 5. Conclusions

Chest RRG poses a challenge due to the limited data availability and the diverse nature of medical reports and variations in pathology expressions. Four transformer-based models were analyzed, highlighting the encoder as the key component, with SwinB being the best choice. This transformer encoder has not been previously explored in RRG. Regarding the decoder, a word-based approach trains faster and achieves more competitive results than a subword-based one.

Techniques like image data augmentation, hard negative mining, and TA were introduced. TA was shown to be an effective method to improve generalization and the quality of the generated reports regarding variability, quality, factual correctness, and completeness, which yields a model that outperforms the state of the art. Moreover, this approach paves the way for new TA approaches based on augmentation with more complex NLP techniques, such as LLM models.

Additionally, NLG-oriented metrics may not be optimal for measuring report quality, as they compare sentences and words without considering semantic meaning or alternative expressions of the same diagnosis. Thus, metrics based on NLP models specialized in chest radiology reports like F_1_cXb and F_1_RG_*ER*_ appear to be crucial in evaluating report quality.

## Figures and Tables

**Figure 1 bioengineering-11-00351-f001:**
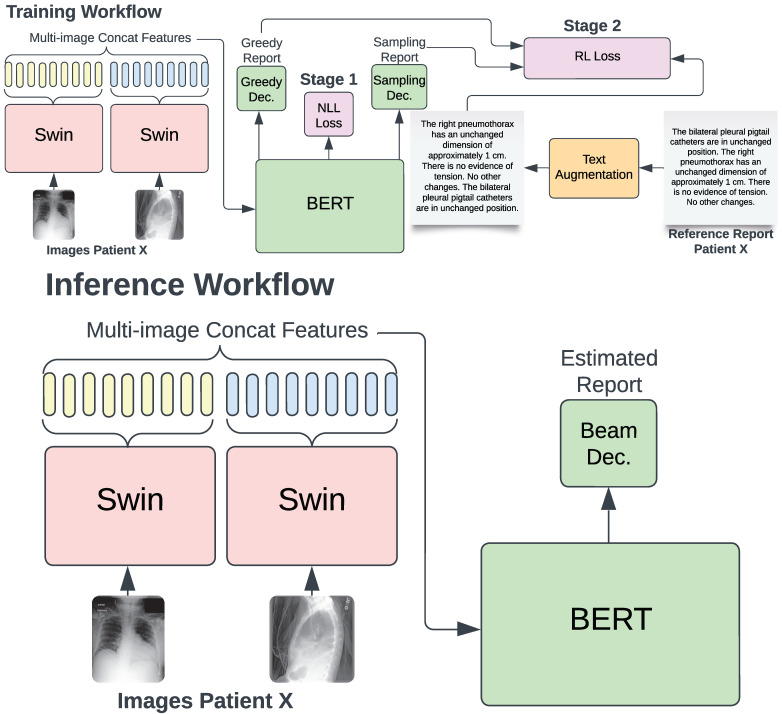
Our training workflow using RL and TA to enhance report quality. In the initial stage, training with NLL employs teacher forcing. Subsequently, during RL training, the SCST algorithm computes rewards utilizing two different reports from two distinct decoding strategies: greedy search and beam-search multinomial sampling. Inference on validation and test sets is conducted using beam search in both stages.

**Figure 2 bioengineering-11-00351-f002:**
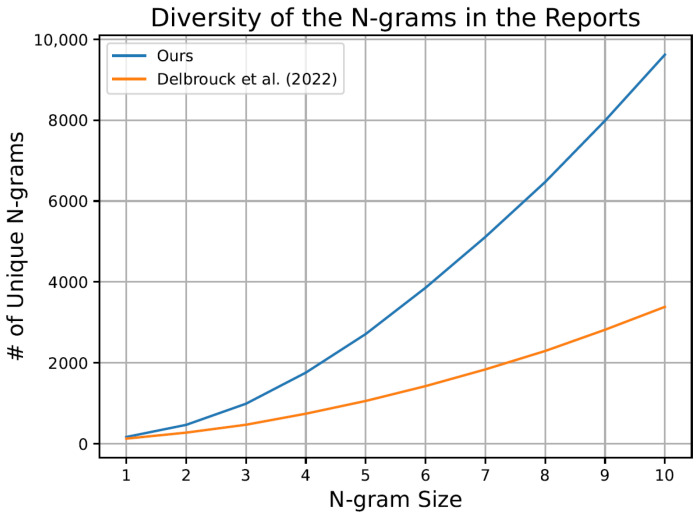
The graph showcases the diversity of N-grams within the generated reports using the MIMIC-CXR test set, encompassing the proposal by Delbrouck et al. [12] and our approach.

**Table 1 bioengineering-11-00351-t001:** Architecture comparison using four report quality metrics on the MIMIC-CXR and Open-i test sets, RL training hours per epoch, and the number of parameters. The metric values are provided in (%). Bold font indicates the best result obtained for each metric.

Model	F_1_cXb	F_1_RG_*ER*_	BLEU4	ROUGE-L	h/epoch	# Params
MIMIC-CXR						
SwinS+BERT9k RL	61.1	34.9	11.5	25.9	**12**	109 M
SwinS+BERT30k RL	61.4	34.3	11.3	25.2	24	130 M
SwinB+BERT9k RL	**63.1**	**35.4**	**11.7**	**26.5**	14	147 M
SwinB+BERT30k RL	62.9	35.1	11.6	**26.5**	30	168 M
Open-i						
SwinS+BERT9k RL	52.8	44.5	14.8	33.7	-	109 M
SwinS+BERT30k RL	52.8	44.1	14.6	33.4	-	130 M
SwinB+BERT9k RL	**54.7**	**45.6**	**15.1**	**34.3**	-	147 M
SwinB+BERT30k RL	54.4	45.3	14.9	34.1	-	168 M

**Table 2 bioengineering-11-00351-t002:** Ablation study of our best model on the MIMIC-CXR test set. The metric values are provided in (%). Bold font indicates the best result obtained for each metric.

Model	F_1_cXb	F_1_RG_*ER*_	BLEU4	ROUGE-L
SwinB+BERT9k NLL	52.3	22.4	10.1	20.5
+ RL	63.1	35.4	11.7	26.5
+ Image Augment.	63.3	35.7	11.9	26.6
+ Hard Neg. Mining	64.2	35.8	12.0	26.8
+ Text Augment.	**66.2**	**37.8**	**13.2**	**27.1**

**Table 3 bioengineering-11-00351-t003:** Comparison of our model, SwinB+BERT9k, against top state-of-the-art models for chest RRG on the MIMIC-CXR test set and Open-i dataset. The metric values are provided in (%). Bold font indicates the best result obtained for each metric.

State of the Art	Chest-RRG-Oriented		NLG-Oriented	
	**F_1_cXb**	**F_1_RG_*ER*_**	**BLEU4**	**ROUGE-L**
**MIMIC-CXR: NLL models**				
Yang et al. [28]	-	-	11.5	28.4
Pan et al. [29]	-	-	11.2	**28.8**
Yang et al. [30]	-	-	11.1	27.4
Zhao et al. [31]	-	-	10.9	27.5
Nicolson et al. [32]	-	-	**12.7**	28.6
Liu et al. [5]	29.2	-	7.6	-
Chen et al. [9]	34.6	-	8.6	27.7
Miura et al. [11]	44.7	-	11.5	-
Chen et al. [10]	40.5	-	10.6	27.8
Delbrouck et al. [12]	44.8	20.2	10.5	25.3
**Ours: NLL stage**	**57.8**	**27.1**	10.3	22.8
**MIMIC-CXR: RL models**				
Miura et al. [11] -(BERTScr+${\mathrm{fact}}_{ENT}$)	56.7	-	11.1	-
Miura et al. [11] -(BERTScr+${\mathrm{fact}}_{ENTNLI}$)	56.7	-	11.4	-
Delbrouck et al. [12] -(BERTScr+F_1_RG_*ER*_)	62.2	34.7	11.4	26.5
**Ours: RL stage -(BERTScr+F_1_RG_*ER*_)**	**66.2**	**37.8**	**13.2**	**27.1**
**Open-i: NLL models**				
Miura et al. [11]	32.2	-	**12.1**	28.8
Chen et al. [10]	-	-	12.0	**29.8**
Alfarghaly et al. [26]	-	-	11.1	28.9
Donahue et al. [49]	-	-	9.9	27.8
Delbrouck et al. [12]	33.1	26.4	11.4	-
**Ours: NLL stage**	**46.6**	**34.5**	11.2	23.4
**Open-i: RL models**				
Miura et al. [11] -(BERTScr+${\mathrm{fact}}_{ENT}$)	48.3	-	12.0	-
Miura et al. [11] -(BERTScr+${\mathrm{fact}}_{ENTNLI}$)	47.8	-	13.1	-
Delbrouck et al. [12] -(BERTScr+F_1_RG_*ER*_)	49.1	41.2	13.9	32.7
**Ours: RL stage -(BERTScr+F_1_RG_*ER*_)**	**54.7**	**45.6**	**15.1**	**34.3**

**Table 4 bioengineering-11-00351-t004:** Comparison of reports generated by our SwinB+BERT9k model trained with RL and TA against reference reports on the MIMIC-CXR test set. The first and third cases involve a patient whose study contains two images, while the second involves a single image.

Images	Hypothesis Report	Reference Report
Input images = 2	F_1_RG_*ER*_ 63.6 %	
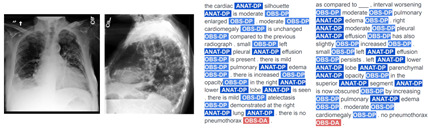		
Input images = 1	F_1_RG_*ER*_ 43.2 %	
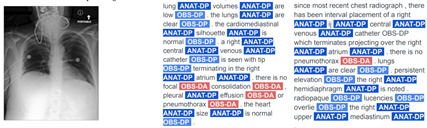		
Input images = 2	F_1_RG_*ER*_ 27.6 %	
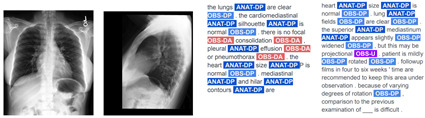		

## Data Availability

The data used in this study are publicly available.

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
