# Peer review of "Improving Radiology Report Generation Quality and Diversity through Reinforcement Learning and Text Augmentation"

_bioengineering, 2024, doi:10.3390/bioengineering11040351_

Round 1

Reviewer 1 Report

Comments and Suggestions for Authors

This study explores the use of deep learning models, such as Swin Transformer and BERT, for generating chest radiology reports. It demonstrates that models like the SwinB encoder with a word-based decoder can produce superior results, especially when combined with techniques like data augmentation and teacher forcing. 

The article is well written, however, some potential issues or limitations can be identified:

  1. The Open-i dataset used for testing contains only 3.3k reports, which may limit the generalizability of the findings compared to larger datasets.

  2. The models in the study were specifically designed to generate the findings section of the reports, excluding samples with empty findings sections. This narrow focus may not fully represent the complexity of generating complete radiology reports.

  3. Evaluation Metrics: While the study employs various metrics like BLEU4, ROUGE-L, F1CheXbert, and F1RGER to evaluate the quality of generated reports, the authors acknowledge that NLG-oriented metrics may not capture semantic meaning effectively. This suggests a potential limitation in assessing the true quality of the generated reports.

  4. The article highlights the necessity for new metrics and strategies to address issues of monotony and improve the evaluation of medical text generation tasks in the future. This indicates a recognition of current limitations in existing evaluation methods.

  5. The article lacks a discussion on potential biases or limitations of the study, such as the generalizability of the findings beyond the specific dataset used.

  6. Additionally, more detailed information on the methodology, specifically the specific TA techniques used and their implementation, would enhance the reproducibility and transparency of the study

Author Response

Thank you for dedicating your time to evaluate this manuscript. Below, you will find comprehensive responses along with highlighted revisions/corrections made accordingly. We appreciate your feedback and suggestions.

Comment 1:

The Open-i dataset used for testing contains only 3.3k reports, which may limit the generalizability of the findings compared to larger datasets.

Response 1:

In the chest RRG task, the two most widely used databases are MIMIC-CXR and Open-i. Due to the limited size of Open-i, it is typically utilized exclusively for testing purposes, as we have done in this work. MIMIC-CXR data originates from the Beth Israel Deaconess Medical Center (BIDMC) in Boston, Massachusetts, while Open-i is a compilation of reports from around the world. Given the highly competitive results obtained with MIMIC-CXR and Open-i, we are confident in the generalizability of our approach.

Comment 2:

The models in the study were specifically designed to generate the findings section of the reports, excluding samples with empty findings sections. This narrow focus may not fully represent the complexity of generating complete radiology reports.

Response 2:

Excluding samples with empty findings sections is the protocol adopted by the other works in the state of the art [1,2]. We have followed this protocol to ensure a fair comparison with other state-of-the-art proposals.

[1] Miura, Yasuhide, et al. "Improving factual completeness and consistency of image-to-text radiology report generation." arXiv preprint arXiv:2010.10042 (2020).

[2] Delbrouck, Jean-Benoit, et al. "Improving the factual correctness of radiology report generation with semantic rewards." arXiv preprint arXiv:2210.12186 (2022).

Comment 3:

Evaluation Metrics: While the study employs various metrics like BLEU4, ROUGE-L, F1CheXbert, and F1RGER to evaluate the quality of generated reports, the authors acknowledge that NLG-oriented metrics may not capture semantic meaning effectively. This suggests a potential limitation in assessing the true quality of the generated reports.

The article highlights the necessity for new metrics and strategies to address issues of monotony and improve the evaluation of medical text generation tasks in the future. This indicates a recognition of current limitations in existing evaluation methods.

Response 3:

Evaluating the true quality of the generated reports is challenging and a current research topic. To date, chest RRG-oriented metrics are the best option, as they focus on the presence of pathologies mentioned in the reference report and their existence, absence, or uncertainty, as well as their relationships and entities in the reports.

Furthermore, we have decided to incorporate in the Discussion a method for measuring the diversity of the reports generated by our model compared to the best model in the state of the art (Delbrouck et al.) (Figure 2 and Lin. 355 to 363)

Given the complexity of this task, we suggest as a future research direction the utilization of Large Language Models to evaluate coherence, factual correctness, and completeness (Discussion: Lin. 388 to 389 and Conclusion: Lin. 434 to 436).

Comment 4: 

The article lacks a discussion on potential biases or limitations of the study, such as the generalizability of the findings beyond the specific dataset used.

Response 4: 

We find this comment highly insightful, prompting us to enhance the Discussion by addressing various limitations of our approach, such as multi-language support, lack of standardization in reports, diverse reporting styles across institutions, and improvements in Picture Archiving and Communication System (PACS) processes (Lin. 390 to 404). Furthermore, we have added a paragraph discussing the application of our approach to different radiological scenarios (Lin. 412 to 423)

Regarding generalizability, we demonstrated our approach's robustness by employing Open-i, an external dataset distinct from MIMIC-CXR, for testing purposes. This strategy yielded the most competitive results in the state of the art.

Comment 5:

Additionally, more detailed information on the methodology, specifically the specific TA techniques used and their implementation, would enhance the reproducibility and transparency of the study

Response 5:

We have expanded the methodology and provided a more detailed explanation of the RL and TA stages to enhance the text and facilitate understanding (Lin. 152 to 172, and Equation 1, and Lin. 191 to 195). As for reproducibility and transparency, we are committed to openly sharing our code on GitHub, making it accessible to everyone.

Reviewer 2 Report

Comments and Suggestions for Authors

The manuscript titled presents an innovative approach to Radiology Report Generation (RRG) utilizing a Vision Encoder-Decoder (VED) framework. This study introduces reinforcement learning and text augmentation techniques to address the limitations of traditional methods in generating diverse and generalized medical reports from radiographic images. While the research introduces significant advancements in RRG, particularly in enhancing report quality and variability, there are several areas where the manuscript could benefit from further elaboration and comparison with existing literature.

The manuscript outlines a two-stage VED architecture employing conventional negative log-likelihood (NLL) training followed by reinforcement learning (RL) optimization. However, the clarity of presentation, especially regarding the implementation details of the RL optimization stage and the integration of text augmentation strategies, could be improved. Providing more granular information on the algorithmic processes, parameter selection, and the rationale behind the design choices would enhance readers' understanding and the reproducibility of the work.

The study would benefit from a more comprehensive comparison with existing methods in RRG, particularly those leveraging advanced computational techniques. Incorporating discussions around works such as "Heterogeneous Network Representation Learning Approach for Ethereum Identity Identification" and "Online Policy Learning-Based Output-Feedback Optimal Control of Continuous-Time Systems" could provide valuable insights into the manuscript's context. While these studies are not directly related to RRG, their methodologies in handling complex data and ensuring interactive safety could offer valuable parallels or contrasts to the current work, enriching the discussion of the manuscript's contributions.

 The manuscript details experiments conducted to validate the proposed method's performance. To bolster the findings, a more extensive comparative analysis involving other existing systems or techniques under similar conditions could provide a clearer demonstration of the proposed system's superiority. This could involve metrics such as report diversity, factual accuracy, and adaptability to different radiological scenarios.

 Expanding the discussion on the practical implications of the findings, especially in real-world RRG scenarios, would significantly enhance the manuscript's relevance. Addressing scalability, adaptability to different radiological contexts, and the integration of such systems into existing healthcare frameworks could provide actionable insights for practitioners and researchers. Furthermore, outlining specific future research directions that build on the current findings, such as exploring the effects of different augmentation techniques or investigating the long-term impact of RL on RRG, would provide a clearer roadmap for advancing the field.

By addressing these concerns, the manuscript could significantly strengthen its technical depth, clarity, and relevance, providing a clearer insight into the proposed system's novelty, effectiveness, and potential applications in the broader context of medical imaging and report generation.

Author Response

Thank you for dedicating your time to evaluate this manuscript. Below, you will find comprehensive responses along with highlighted revisions/corrections made accordingly. We appreciate your feedback and suggestions.

Comment 1:

However, the clarity of presentation, especially regarding the implementation details of the RL optimization stage and the integration of text augmentation strategies, could be improved.

Response 1:

We agree with the comment and have expanded and elaborated in detail on the RL stage and its corresponding mathematical formulation. Additionally, we have outlined how rewards are calculated and how loss is obtained to train the VED model (Lin. 152 to 172, and Equation 1). We have also clarified the explanation of TA and how it is incorporated into the training workflow (Lin. 191 to 195).

Comment 2:

The study would benefit from a more comprehensive comparison with existing methods in RRG, particularly those leveraging advanced computational techniques. 

Response 2:

RRG is a cutting-edge task that has been predominantly tackled by deep learning in recent years. The most competitive proposals leverage neural networks and have been included in Table 3 for comparison with our approach. We have employed widely-used metrics from the state of the art to analyze the competitiveness of our approach. In response to this request, we have included in the Discussion a paragraph and a figure to illustrate how our reports exhibit greater diversity and less monotony compared to the best proposal in the state of the art (Delbrouck et al.) (Figure 2 and Lin. 355 to 363).

Comment 3:

Incorporating discussions around works such as "Heterogeneous Network Representation Learning Approach for Ethereum Identity Identification" and "Online Policy Learning-Based Output-Feedback Optimal Control of Continuous-Time Systems" could provide valuable insights into the manuscript's context.

Response 3:

Thank you very much for recommending these works; we have incorporated them to enhance the introduction and provide context to the paper. These two investigations are similar to our complex data context and its sensitivity.

Comment 4:

 The manuscript details experiments conducted to validate the proposed method's performance. To bolster the findings, a more extensive comparative analysis involving other existing systems or techniques under similar conditions could provide a clearer demonstration of the proposed system's superiority. This could involve metrics such as report diversity, factual accuracy, and adaptability to different radiological scenarios.

Response 4:

As RRG is a recent task mainly addressed by deep learning, there are few existing works, and those featured in Table 3 represent the latest and most effective approaches. Furthermore, as state-of-the-art proposals often do not release their code, conducting such an analysis poses significant challenges. Hence, we are committed to sharing our code on GitHub to enhance the reproducibility of our approach.

* Factual accuracy:

Regarding metrics for factual accuracy, it is worth noting that F1cXb performs this task at the level of present, absent, and uncertain pathologies. Meanwhile, F1RGER evaluates whether the entities present in the reference report and the relationships therein also exist in the reports generated by our model. Therefore, the analysis conducted is at the level of factual accuracy using the F1-score (Table 3)

* Diversity:

Regarding report diversity, we assert that our reports achieve greater diversity and lower monotony than others, as the papers of the proposals we compare mention repetition and monotony in most of the generated reports.

We have decided to incorporate in the Discussion a method for measuring the diversity of the reports generated by our model compared to the best model in the state of the art (Delbrouck et al.) (Figure 2, Lin. 355 to 363). The Figure demonstrates, through n-grams, how our model generates more diverse reports than Delbrouck et al. Additionally, our model's tendency to produce greater diversity shows an average growth nearly three times higher.

* Adaptability to different radiological scenarios:

Due to the recent emergence of the RRG task and the need for large amounts of data to train models for this task, typically, only the chest radiology scenario is covered. This is because chest radiographs have the largest databases available. Regarding adaptability to different radiological scenarios, our approach is flexible and independent, capable of being applied to other radiological scenarios. Our neural model requires no modifications to process images or reports from other body regions, and our training workflow and Text Augmentation technique remain unchanged. The only adjustment necessary would be the utilization of a different metric to optimize during the RL stage, as F1RadGraph is tailored explicitly for chest radiographs. Thus, we believe it could be an interesting avenue for future research, and we included it in the Discussion (Lin. 412 to 423)

Comment 5:

Expanding the discussion on the practical implications of the findings, especially in real-world RRG scenarios, would significantly enhance the manuscript's relevance. Addressing scalability, adaptability to different radiological contexts, and the integration of such systems into existing healthcare frameworks could provide actionable insights for practitioners and researchers.

Response 5:

We have extended the Discussion section to accommodate the significance of this comment. In doing so, we have addressed potential biases or limitations in our approach. These include considerations such as multi-language support, the absence of standardized reporting protocols, variations in reporting styles among different institutions, enhancements in Picture Archiving and Communication System (PACS) processes tailored for expert radiologists, and the adaptability of our approach to diverse radiological contexts (Lines 390 to 404, and 412 to 423). Furthermore, we have delineated the minimal hardware requisites conforming to contemporary standards for implementing our approach. (Lines 405 to 411).

Comment 6:

Furthermore, outlining specific future research directions that build on the current findings, such as exploring the effects of different augmentation techniques or investigating the long-term impact of RL on RRG, would provide a clearer roadmap for advancing the field.

Response 6:

Due to the promising results obtained, in both the Discussion and Conclusion sections, we elaborated on how utilizing Large Language Models (LLMs) could represent one of the most intriguing directions for advancing in this field. LLMs can be instrumental in developing more advanced Text Augmentation (TA) techniques. They can also offer robust semantic and syntactic evaluations to facilitate the development of new Reinforcement Learning (RL) rewards (Discussion: Lin. 388 to 389 and Conclusion: Lin. 434 to 436).

Reviewer 3 Report

Comments and Suggestions for Authors

Reinforcement learning and text augmentation were applied to enhance radiology report generation quality and diversity. There are some key comments to be addressed before a recommendation of acceptance. Please refer to my comments below:
Comment 1. Abstract: Share the research results and implications.
Comment 2. Keywords: More terms should be included to better reflect the scope of the paper.
Comment 3. Section 1 Introduction:
(a) The importance of the research topic should be strengthened, particularly on the usefulness of radiology reports.
(b) Rewrite Subsection 1.1 Related works. Please provide a good summary of the methodologies, results, and limitations of the latest works (mainly the recent 5-year journal articles).
(c) Clearly state the research contributions of the paper.
Comment 4. Section 2 Materials and Methods:
(a) Add an introductory paragraph before Subsection 2.1.
(b) Figure 1 does not fully include all building blocks for the model.
(c) Methodology lacks details. Include equations, pseudo-code, flow chart, etc., where appropriate.
Comment 5. Section 3 Results:
(a) Provide a detailed analysis of the hyper-parameters tuning.
(b) For the models being compared, include in-text citations.
(c) Conduct an ablation study.
(d) Compare your works with the latest existing works.
Comment 6. How can you confirm the effectiveness of the generated reports? Did you consult the advice from radiologists or medical doctors?

Comments on the Quality of English Language

There are some minor typos and formatting issues.

Author Response

Thank you for dedicating your time to evaluate this manuscript. Below, you will find comprehensive responses along with highlighted revisions/corrections made accordingly. We appreciate your feedback and suggestions.

Comment 1:

Abstract: Share the research results and implications.

Response 1:

We have updated the abstract to incorporate the results and emphasize their implications (Lin. 8 to 10).

Comment 2:

Keywords: More terms should be included to better reflect the scope of the paper.

Response 2:

We have included the terms "Machine Learning", "Chest X-rays," "Medical Image," and "Text Generation" (Lin. 13 to 14).

Comment 3:

Section 1 Introduction:

(a) The importance of the research topic should be strengthened, particularly on the usefulness of radiology reports.

(b) Rewrite Subsection 1.1 Related works. Please provide a good summary of the methodologies, results, and limitations of the latest works (mainly the recent 5-year journal articles).

(c) Clearly state the research contributions of the paper.

Response 3:

  1. We have reinforced the RRG task's importance in the introduction's first paragraph (Lin. 21 to 27).
  2. Since the RRG task is relatively recent, most relevant proposals and approaches have been included in the initial version of this manuscript. In order to enhance the Related Work section, we have addressed the request by adding two new paragraphs with new additional references focused on recent 5-year journal articles (Lin. 63 to 72, and Lin. 73 to 90).
  3. We have explicitly elaborated on the paper's contributions in the introduction's first subsection (Lin. 38 to 47).

Comment 4:

Section 2 Materials and Methods:

(a) Add an introductory paragraph before Subsection 2.1.

(b) Figure 1 does not fully include all building blocks for the model.

(c) Methodology lacks details. Include equations, pseudo-code, flow chart, etc., where appropriate.

Response 4:

  1. We have added an introductory paragraph before Subsection 2.1 (Lin. 112 to 117).
  2. Figure 1 encompasses all the necessary blocks to comprehend the proposed approach. The Swin [1] and BERT [2] models are two widely recognized and large-scale deep learning models extensively employed in tasks related to computer vision and natural language processing. Given their prominence and proper referencing, we deemed it unnecessary to detail every characteristic of these models intricately. Moreover, we have updated Figure 1 to delineate the training and inference workflows to enhance comprehension.
  3. We have enhanced the methodology by providing more detail on the RL stage and the equation necessary for calculating rewards (Lin. 152 to 172, and Equation 1). Together with Figure 1, these additions should be sufficient to understand the workflow of the proposed RRG system. The Text Augmentation methodology has been further elaborated to enhance the explanation of the training workflow (Lin. 191 to 195). Furthermore, it is worth noting that additional explanations or pseudo-code are unnecessary as we intend to make the code publicly available on GitHub.

[1] Liu, Ze, et al. "Swin transformer: Hierarchical vision transformer using shifted windows." Proceedings of the IEEE/CVF international conference on computer vision. 2021.

[2] Devlin, Jacob, et al. "Bert: Pre-training of deep bidirectional transformers for language understanding." arXiv preprint arXiv:1810.04805 (2018).

Comment 5:

Section 3 Results:

(a) Provide a detailed analysis of the hyper-parameters tuning.

(b) For the models being compared, include in-text citations.

(c) Conduct an ablation study.

(d) Compare your works with the latest existing works.

Response 5:

  1. The hyper-parameter tuning presented corresponds to the official Swin paper [1] for both the base architecture (SwinB) and the small variant (SwinS). We opted not to explore different hyper-parameters, instead focusing on these two architectures, which we deemed most suitable for RRG. As for BERT, we employed the original BERT model from the paper [2], without conducting any parameter analysis. We solely modified BERT to reduce its size by utilizing 3 layers, a common practice in various tasks due to its large scale. The 50k vocabulary aligns with BERT's original subword-based vocabulary, while the 9k vocabulary corresponds to words found in the report vocabulary. The only hyper-parameter exploration conducted was to find the optimal learning rate for training using the MIMIC-CXR validation set. As outlined in the paper, we performed this tuning through grid search, testing 12 different values using the validation set of MIMIC-CXR (Lin. 243 to 247). Regarding the weights used to compute the RL Loss, as we optimize for 2 metrics (BERTScore and F1RGER) and consider it essential to still account for the NLL Loss, we have weighted the NLL Loss by 0.010 and BERTScore and F1RGER by 0.495 each.
  2. Resolved.
  3. The ablation study is presented in Table 2.
  4. In Table 3, we present a comparative analysis of our approach alongside other state-of-the-art models. We have included seven recent models from journal articles published within the last five years. (these seven models are highlighted in blue). Additionally, we have rewritten the comparison of the models and added a paragraph about the comparison (Lin 294 to 304, and Lin 318 to 334). Finding relevant research papers is challenging due to two main factors. The first is the novelty of the task, and the second is the traditional reliance on NLG-oriented metrics such as BLEU4 and ROUGE-L for assessing system quality. However, these metrics do not account for pathologies' presence, absence, or uncertainty, nor do they consider the relationships between them in the report. To address these limitations, we employ chest RRG-oriented metrics. Nonetheless, these metrics are relatively new, making them difficult to locate in journal literature.

[1] Liu, Ze, et al. "Swin transformer: Hierarchical vision transformer using shifted windows." Proceedings of the IEEE/CVF international conference on computer vision. 2021.

[2] Devlin, Jacob, et al. "Bert: Pre-training of deep bidirectional transformers for language understanding." arXiv preprint arXiv:1810.04805 (2018).

Comment 6:

How can you confirm the effectiveness of the generated reports? Did you consult the advice from radiologists or medical doctors?

Response 6:

Chest RRG-oriented metrics emerge as a crucial tool. These metrics measure the quality, factual accuracy, and relationships of present, absent, or uncertain pathologies in the reports using neural language models. Consequently, we can assess the quality and effectiveness of the obtained reports. Currently, the majority of proposals for the RRG task do not employ teams of expert radiologists or medical doctors due to the significant investment required. As a future direction, we aim to enhance our workflow with the assistance of expert radiologists and the introduction of new chest RRG-oriented metrics.

Round 2

Reviewer 1 Report

Comments and Suggestions for Authors

The authors have incorporated the comments and also given valid justification. 

No more changes are required in the article and from my side, it is recommended for publication.

Author Response

Thank you for dedicating your time to evaluate this manuscript. We appreciated your feedback and suggestions.

Reviewer 2 Report

Comments and Suggestions for Authors

The authors have addressed all of my concerns. The current version can be accepted.

Author Response

(The authors gave the same response as above.)

Reviewer 3 Report

Comments and Suggestions for Authors

The authors have significantly enhanced the paper quality. I have some minor follow-up comments.
Follow-up comment 1. The four research contributions are not precise. Particularly, introducing a method and acknowledging the constraints are not proper descriptions. Ensure proper action verbs are used and technical content is presented.
Follow-up comment 2. Subsection 1.1 should elaborate on the limitations of the cited works.
Follow-up comment 3. Figure 2, in-text citation is needed for the existing work.
Follow-up comment 4. Why were [51] and [52] cited in the conclusion?

Author Response

Thank you for dedicating your time to evaluate this manuscript. Below, you will find comprehensive responses along with highlighted revisions/corrections made accordingly. We appreciate your feedback and suggestions.

Comment 1:

Follow-up comment 1. The four research contributions are not precise. Particularly, introducing a method and acknowledging the constraints are not proper descriptions. Ensure proper action verbs are used and technical content is presented.

Response 1:

We have enhanced the contributions in response to the feedback received.

Comment 2:

Follow-up comment 2. Subsection 1.1 should elaborate on the limitations of the cited works.

Response 2:

In response to this request, we have incorporated the limitations of the presented works into two separate sections (Lines 91 to 96 and Lines 116 to 119). In these sections, we articulate that the proposals primarily focus on comparing with other models using metrics such as BLEU or ROUGE-L, which may not be entirely suitable for measuring pathologies' presence, absence, or uncertainty. Furthermore, these metrics fail to consider the interrelation of pathologies within the report. Additionally, we emphasize the repetition and monotony observed in the generated reports of these proposals.

Comment 3:

Follow-up comment 3. Figure 2, in-text citation is needed for the existing work.

Response 3:

Resolved.

Comment 4:

Follow-up comment 4. Why were [51] and [52] cited in the conclusion?

Response 4:

Considering this comment, we believe it is better to include these citations in the Discussion rather than the Conclusion. As a result, we have modified Line 395 to 396 of the Discussion and Line 441 to 443 of the Conclusion accordingly.